# Population Genomics, Transcriptional Response to Heat Shock, and Gut Microbiota of the Hong Kong Oyster *Magallana hongkongensis*

Yichun Xie [1,†] , Elaine Y. Y. Huang [1,†] , Wenyan Nong [1,†], Sean T. S. Law [1,†], Yifei Yu [1,†], Khan Cheung [2,†],
Yiqian Li [1], Cheuk Fung Wong [1], Ho Yin Yip [1], Patrick W. S. Joyce [3] , King Ming Chan [3] , Ka Hou Chu [3,4] ,
Bayden D. Russell [2,*] , Laura J. Falkenberg [3,*] and Jerome H. L. Hui [1,*]

1 State Key Laboratory of Agrobiotechnology, Simon F.S. Li Marine Science Laboratory, School of Life Sciences,
The Chinese University of Hong Kong, Hong Kong 999077, China; xieyichun50@link.cuhk.edu.hk (Y.X.);
hyyalien@gmail.com (E.Y.Y.H.); nongwenyan@cuhk.edu.hk (W.N.); lolhahasean@gmail.com (S.T.S.L.);
yyf9701@gmail.com (Y.Y.); liyiqian1994@gmail.com (Y.L.); wcfung14@gmail.com (C.F.W.);
yhy12@cuhk.edu.hk (H.Y.Y.)
2 The Swire Institute of Marine Science, School of Biological Sciences, The University of Hong Kong,
Hong Kong 999077, China; khancheung@gmail.com
3 Simon F.S. Li Marine Science Laboratory, School of Life Sciences, The Chinese University of Hong Kong,
Hong Kong 999077, China; patrickjoyce@cuhk.edu.hk (P.W.S.J.); chankingming@gmail.com (K.M.C.);
kahouchu@cuhk.edu.hk (K.H.C.)
4 Southern Marine Science and Engineering Guangdong Laboratory (Guangzhou), Guangzhou 510301, China
* Correspondence: brussell@hku.hk (B.D.R.); laurafalkenberg@cuhk.edu.hk (L.J.F.);
jeromehui@cuhk.edu.hk (J.H.L.H.)
† These authors contributed equally to this work.

**Abstract:** The Hong Kong oyster *Magallana hongkongensis*, previously known as *Crassostrea hongkongensis*, is a true oyster species native to the estuarine-coast of the Pearl River Delta in southern China. The species—with scientific, ecological, cultural, and nutritional importance—has been farmed for hundreds of years. However, there is only limited information on its genetics, stress adaptation mechanisms, and gut microbiota, restricting the sustainable production and use of oyster resources. Here, we present population structure analysis on *M. hongkongensis* oysters collected from Deep Bay and Lantau Island in Hong Kong, as well as transcriptome analysis on heat shock responses and the gut microbiota profile of *M. hongkongensis* oysters collected from Deep Bay. Single nucleotide polymorphisms (SNPs), including those on the homeobox genes and heat shock protein genes, were revealed by the whole genome resequencing. Transcriptomes of oysters incubated at 25 °C and 32 °C for 24 h were sequenced which revealed the heat-induced regulation of heat shock protein pathway genes. Furthermore, the gut microbe community was detected by 16S rRNA sequencing which identified Cyanobacteria, Proteobacteria and Spirochaetes as the most abundant phyla. This study reveals the molecular basis for the adaptation of the oyster *M. hongkongensis* to environmental conditions.

**Keywords:** *Magallana hongkongensis*; population structure; heat shock responses; gut microbiome; adaptation

## 1. Introduction

Hong Kong is one of the top seafood consumers per capita in the world [1]. The Hong Kong oyster *Magallana hongkongensis* is a true oyster species inhabiting the estuarine region of the Pearl River Delta and the surrounding coastal areas of southern China [2,3]. The species occurs wild on rocks in estuarine regions and has been farmed for ~700 years in China, becoming one of the most prominent forms of aquaculture activities [4]. In Hong Kong, Lau Fau Shan at Deep Bay (Shenzhen Bay) is the only known locality for *M. hongkongensis* farming [5]. Oysters in Hong Kong were traditionally farmed with bottom

culture, but since the 1970s, raft culture dominates production. The majority of locally cultivated oysters are sold fresh to the Hong Kong market, sun-dried as dried seafood ("golden oysters"), or deep processed as oyster sauce for seasoning. In recent decades, however, climate change has increased risks to the viability of bivalve populations around the world [6,7]; oyster diseases, mortality, and population invasions have increased under such circumstances [8–10]. Moreover, estuarine and coastal regions have undergone a higher rate of urbanisation, warming and ocean acidification, threatening the oyster habitation in these areas [11]. The problems driven by global change can be further complicated by modified local conditions; for example, the increase in oyster disease under a changing climate can be influenced by the presence of antibiotic resistant bacteria [8,12]. Consequently, there is a need for research to provide the scientific basis on oyster restoration and sustainable production. Such research is now possible as chromosome-level genome assemblies of *M. hongkongensis* have recently been released [13,14], providing an opportunity and the resources for in-depth molecular studies.

Genetic diversity is integral to the resistance and resilience of natural populations to changeable environmental conditions [15], with increasing awareness around the importance of incorporating evolutionary considerations into species conservation programs. Meanwhile, a thorough understanding on the baseline ecological, demographical, and genetic characteristics of the target species is the prerequisites of effective interventions [16]. Yet, the population genetics of marine species has been historically understudied. It was previously believed that marine populations were highly interconnected due to high potential for dispersal [17,18]. More recently, studies clarified the local adaptation mechanism of several oysters, with underlying mechanisms including the differences on genotype heterogeneity and nucleotide polymorphism [19–21]. *Magallana* oysters have high fecundity and prolonged pelagic larval durations (two to three weeks) [22], likely providing high gene-flow among populations and reducing the likelihood of local adaptation [23]. Unlike other *Magallana* spp. found along the Chinese coastline, *M. hongkongensis* has a remarkably low sequence diversity in its mitogenome, indicating the possibility of recent population bottlenecks [24]. Nevertheless, the population structure of *M. hongkongensis* remains unclarified.

As sessile animals that inhabit estuarine coastlines, oysters face a changing aquatic environment. Challenges from altered temperature, salinity, toxicants, and microbes threaten the development and survival of oysters. For *Crassostrea virginica*, biomarkers of heat stress and oxidative stress are significantly increased after only one week exposure to the elevated temperature (32 °C), accompanied by a dramatical decrease in sperm production [25]. For the Pacific oyster *Magallana gigas*, the temperature experienced during the larval stage regulates the protein processing genes and adult longevity genes [26] and can inhibit the larval development and immune responses [27]. In the adults, stressors of different types, including temperature, salinity, air exposure, and heavy metals, all activated the defense genes and signal transduction genes [28]. When exposed to various stressors, heat shock proteins and heat shock factors perform a central role in determining the responses [28,29]. In *M. hongkongensis,* very little research has been done on heat stress responses, though digital gene expression tag profiling of gill tissue has identified enriched regulation on protein processing in the endoplasmic reticulum at 37 °C [30]. Therefore, although the general mechanism of thermal tolerance has been investigated in other oyster species, understanding of the specific pattern of global gene expression regulation and gene structure modulation during heat shock response remains scant for populations of *M. hongkongensis* in Hong Kong.

Oysters are filter feeders and ingest organic debris, bacteria, microzooplankton and microalgae from seawater as food resources [31]. Thus, the microbiota of oysters can be shaped according to the environment that the host inhabits [32–35], and previous studies have shown strong correlation between the gut microbial community and environmental conditions [36,37]. Yet, there are indications that there can be host selection in the microbiome, leading to microbiome plasticity, with the microbiota diversity of the pearl oyster

*Pinctada fucata martensii* being lower in the intestine than the surrounding water [38–41]. Further, the gut microbiome also affects oyster health, including the outbreak of oyster diseases [8]. Therefore, understanding the microbiome profile is important for the restoration and production of local oysters, and, yet, the microbiome remains unknown in *M. hongkongensis*.

For the oyster *M. hongkongensis* collected in the Deep Bay and Lantau Island areas of Hong Kong, using the recently published genome [13], this study aims to (i) reveal the population genetic structure, (ii) elucidate the transcriptomic response to heat shock treatment, and (iii) identify the gut microbiota of *M. hongkongensis*.

## 2. Materials and Methods

### 2.1. Samples Collection, DNA Extraction, Species Identification, Sequencing

A total of 44 M. hongkongensis samples were collected from oyster rafts at different localities of Deep Bay (Shenzhen Bay) in May and June 2019 (named as B1-B10 and KM), and five additional samples were collected from intertidal mudflat at Lantau Island in September 2019 (named as HKU) (Figure 1 and Supplementary Table S1). The adductor muscle (0.02 g) was dissected individually for DNA extraction using a Purelink Genomic Mini Kit (Invitrogen, US), following the manufacturer's instructions. The quality and quantity of the extracted DNA was determined by gel electrophoresis with a Gel Doc$^{TM}$ EZ imager (Bio-rad) and NanoDrop (Thermo Fisher), respectively. To confirm the species identity of the collected oysters, the mitochondrial DNA cytochrome oxidase subunit I (COI) gene was amplified by polymerase chain reaction (PCR). The reaction was carried out on a T100$^{TM}$ thermocycler (Bio-Rad) with the following parameters: 1 cycle of the denaturation step at 95 °C for 3 min, 39 cycles for 30 s at 95 °C, 30 s at 56 °C, and 45 s at 72 °C, and a final extension step at 72 °C for 5 min. The total volume of each reaction was 20 μL, including 20–50 ng of DNA template and the final concentration of $1\times$ buffer, 0.8 mM dNTPs, 1.5 mM MgCl$_2$, 0.4 μM of each forward and reverse primer, and 1 unit of Taq DNA polymerase. The universal primers used for target amplification were LCO1490 (5′-GGT CAA CAA ATC ATA AAG ATA TTG G-3′) and HCO2198 (5′-TAA ACT TCA GGG TGA CCA AAA AAT CA-3′) [42]. The amplified product was checked with gel electrophoresis and sent to BGI Hong Kong for Sanger sequencing. The sequencing results were BLAST searched against the NCBI nucleotide database for identity confirmation. A DNA sample of each confirmed *M. hongkongensis* individual was sent to Novogene Company Ltd. for whole genome sequencing (PE150, 350 bp library, with 2 GB raw data each) using Illumina NovaSeq 6000 platform.

### 2.2. Bioinformatic Analyses

The obtained raw sequencing reads were trimmed and processed with trimmomatic (version 0.39 with parameters "ILLUMINACLIP:TruSeq3-PE.fa:2:30:10 SLIDINGWIN-DOW:4:5 LEADING:5 TRAILING:5 MINLEN:25") and Kraken (version 2.0.8) for contamination removal [43,44]. The processed reads were then mapped to the published Hong Kong oyster *M. hongkongensis* genome [13] using a Burrows-Wheeler Aligner (BWA) (version 0.7.12) with "-M" activated (mark shorter split hits as secondary) and other parameters on default values, for further SNPs analysis of single nucleotide polymorphisms (SNPs). PCR duplicates were removed with Picard 'MarkDuplicates'.

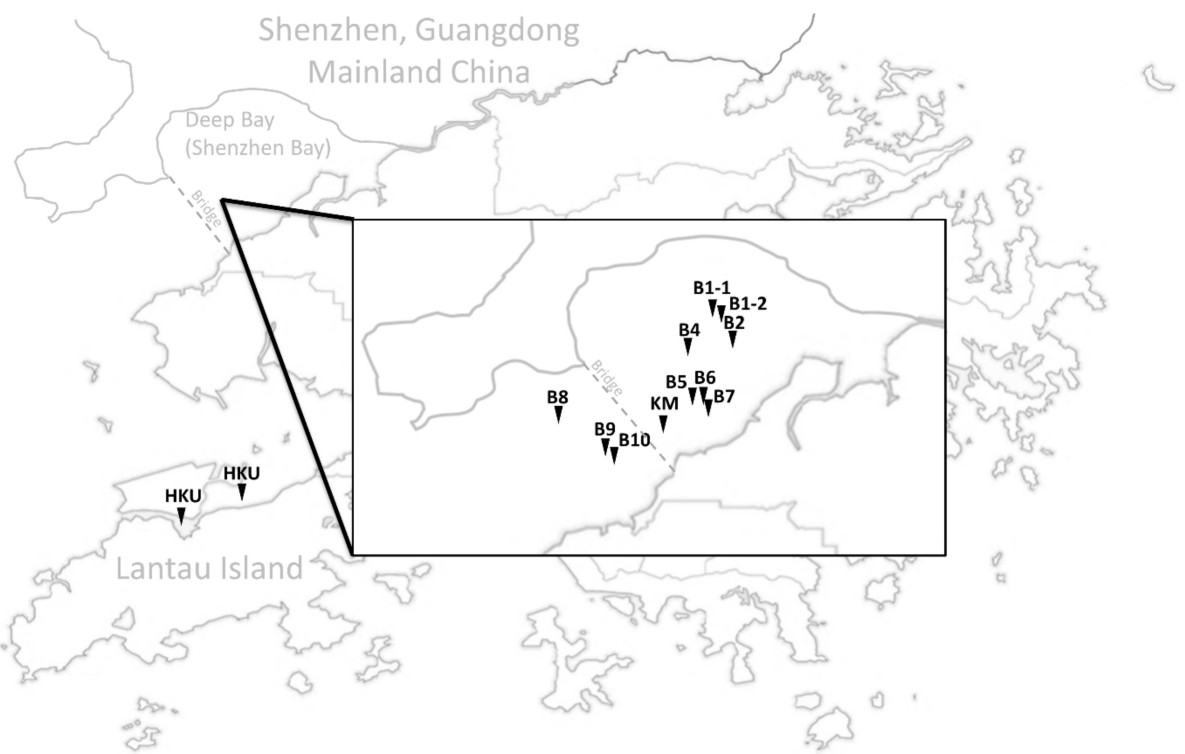

**Figure 1.** Localities of the 49 collected Hong Kong oyster *M. hongkongensis*.

### 2.3. Population Genomic Analyses

SNPs were called with the 'VariantFiltration' function in the Genome Analysis Toolkit (GATK) with hard filtering criteria (–filter-expression "FS > 60.0 || MQ < 40.0 || QD < 2.0 || SOR > 3.0") [45]. The SNP dataset was annotated with the gene models of the reference assembly using SnpEff [46]. The SNP dataset was filtered with criteria of bi-allelic, no missing rate, and excluding annotated repeats ("–min-alleles 2 –max-alleles 2 –max-missing 1 –exclude-bed") using VCFtools [47], followed by linkage disequilibrium pruning "–indep 50 5 2" and minor-allele frequency threshold of 5% "–maf 0.05" using PLINK (v1.9) [48]. Principal component analysis (PCA) was also conducted using the "–pca" option in PLINK to extract eigenvectors and eigenvalues. Subsequently, a neighbor joining (NJ) phylogenetic tree was constructed in R using 'poppr' [49], 'ade4' [50], 'ape' [51], 'vcfR' [52], 'adegenet' [53] packages with 1000 bootstraps and was visualized using Evolview v3 [54].

### 2.4. Annotation of Homeobox Gene and Heat Shock Protein Family

The homeobox gene family was referred to the genomic analysis carried out by Li et al. [13]. Heat shock proteins (HSP) were identified by BLAST search on conserved domains. A total of 202 HSP genes was identified (Table 1, detailed in Supplementary Materials, Table S2). SNPs at Homeobox genes and heat shock proteins were annotated as described in the previous section. The SNPs with the minor allele frequency (MAFs) less than 5% or the missing rate of a genotype for the population over 10% were discarded. Selected SNPs were also checked by PCR amplification and Sanger sequencing as described above, and primer sequences were listed in Supplementary Materials, Table S3. Those 44 individuals collected from Shenzhen Bay were further pooled into one batch, and compared against the five individuals collected from Lantau Island. SNPs or INDELs presented in at least 50% of individuals in the group and found in no more than 1 sample of another group are considered as variants unique to the specific population.

**Table 1.** Number of heat shock protein (Hsp) families for the heat shock response analysis.

| Hsp Families | Number of Blasted Genes (n) |
|---|---|
| HSPE | 2 |
| HSPB | 20 |
| HSP40 | 50 |
| HSP70 | 123 |
| HSPC(HSP90) | 5 |
| HSP110 | 2 |

*2.5. Heat Shock Treatment and RNA-seq*

Four *M. hongkongensis* purchased from Deep Bay in March 2021 were used for the heat shock experiment. Each oyster was separately placed into a 5 L tank with sand-filtered seawater for a week of acclimation at 25 °C, while the temperature and salinity were under monitored. The water temperature for two of the acclimated oysters were increased to 32 °C for 24 hours as the acute heat shock group, whereas the other two oysters remained at 25 °C as the control group. Both DNA and RNA were extracted from the muscle tissue after the 24-hour treatment. Genomic DNA extraction and species identification procedures were carried out as described above. Total RNA was isolated with TRIzol reagent (Ambion, Austin, USA) following the manufacturer's instructions. The quality and quantity of RNA samples was confirmed with gel electrophoresis and NanoDrop. The RNA samples were stored at −80 °C and sent to Novogene Company Ltd. for transcriptome sequencing with NovaSeq6000 on PE150 strand-specific mode. Six GB of raw data was collected for each sample.

*2.6. Gene Model Prediction and Transcriptome Profiling*

Gene model prediction on the *M. hongkongensis* isolate MH-2019 (GenBank: WFKH00000000.1) was performed as previously described [13]. Functional annotation of the predicted proteome was done using eggNOG [55], with "–pident 40 –seed_ortholog_evalue 0.001 –seed_ortholog_score 60 –query_cover 20 –subject_cover 20 –target_taxa 33,213 – excluded_taxa 7742" to improve the accuracy.

Transcriptomes were then profiled referring to the gene model as follows. Clean reads were processed with FastQC for quality control (Babraham Bioinformatics, The Babraham Institute, UK). After read alignment to the reference genome using HISAT2 (version 2.2.1) [56], gene expression levels were calculated using Stringtie (version 2.1.5) [57]. Only genes with a count-per-million (CPM) over 1 in both replicates were proceed to the following analyses. The generated count matrix was used in edgeR [58] for the expression analysis, and presented in terms of the Trimmed Mean of M-values (TMM). Only the genes with a threshold of false discovery rates (FDR) adjusted $p$-value < 0.05 and $|\log2(\text{fold change})| > 1$ were considered as differentially expressed genes. Functional enrichment analyses were carried out using 'compareCluster()' function in R package 'clusterprofiler' [59], with 'pvalueCutoff = 0.2, pAdjustMethod = "BH", qvalueCutoff = 0.2". Results were visualized using ggplot2 [60].

Alternative splicing (AS) events were identified using CASH (version 2.2.0), a self-construct alternative splicing detector [61]. After sequencing read alignment, duplicated reads were removed using the 'MarkDuplicates' function in Picard tools (version 2.26.5, https://broadinstitute.github.io/picard/, accessed on 1 December 2021). The mapping file in BAM format and the gene model annotation file in GFF3 format were then supplied into CASH. AS events were filtered with the cut-off of $p$-value < 0.05 and false discovery rate using the Benjamini–Hochberg method < 0.05. AS sites were manually checked in Integrative Genomics Viewer (IGV) [62].

*2.7. Gut Microbiota Analysis*

The gut of three *M. hongkongensis* individuals purchased from Deep Bay in October 2019 were dissected, and DNA was extracted from ~10 mg of feces samples using Purelink

Genomic Mini Kit (Invitrogen, US) according to the manufacturer's instructions. Species identification procedures were carried out as described above. Extracted fecal DNA was sent to Novogene Company Ltd. for bacteria amplicon sequencing (16S V3–V4 region with 100 k raw tags each sample) using the Illumina NovaSeq PE250 mode. Operational taxonomic units (OTUs) were classified using QIIME2 with standard workflow [63]. Only assigned OTUs with read count over 10 were further analysed.

### 2.8. Data Availability

Sequencing data generated in this study has been submitted to NCBI under the Bio-Project PRJNA576886. Data summary of population genomic analyses and transcriptomic analyses were listed in Supplementary Materials, Tables S4 and S5, respectively. Genome annotation files were deposited in Figshare https://doi.org/10.6084/m9.figshare.17696531, accessed on 29 December 2021.

### 3. Results

### 3.1. Population Relationships of Local Oysters

A total of 49 oyster individuals was confirmed with the species identity of *M. hongkongensis*. After sequencing the whole genome region and aligning the sequences against the reference genome, 30,022 SNPs were identified in the population with the minor allele frequency no less than 5%. PCA did not suggest significant genetic differences of the collected samples (Figure 2a). The filtered SNPs were then used for phylogenetic tree construction. Oysters collected from different locations were mixed on the Neighbour-joining tree (Figure 2b).

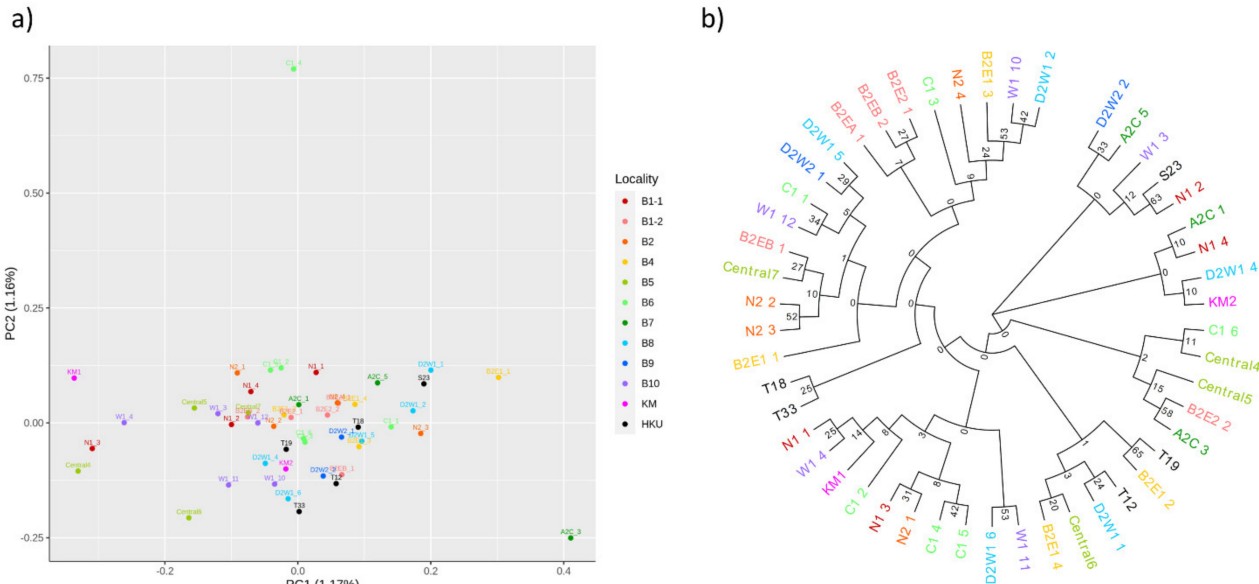

**Figure 2.** Population relationships of the collected *M. hongkongensis* individuals. (**a**) Principal component analysis based on genetic differences. (**b**) Neighbour-joining tree of individuals based on filtered SNPs.

### 3.2. SNPs and INDELs in Homeobox Genes and Heat Shock Protein Genes

Considering the important roles of homeobox (Hox) genes in development and heat shock protein (HSP) genes in stress response, we further analysed the SNPs in the homeobox, Hsp70, and Hsp110 families in the collected *M. hongkongensis* samples. A total of 103 SNPs and 8 INDELs were first identified in Hox genes, and most of them were predicted to result in missense mutations (Supplementary Table S6). Thirty-eight genetic variants in homeobox genes were identified as specific to oysters other than the Lantau Island samples (Supplementary Tables S9 and S10).

In the Hsp gene families, SNP was not found and 693 INDELs were identified (Supplementary Table S7). Among these INDELs, five of them located on three Hsp70 12 A were found to be unique to the specific population (Supplementary Table S8). These INDELs all result in the shift of the open reading frame and change the protein sequences.

### 3.3. Transcriptome Profile in Response to Heat Shock

A total of 36,239 gene loci with 40,086 proteins was predicted in the genome (Table 2). When performing the ortholog assignment and functional annotation analysis against the egg NOG database [55,64], 14,663 genes were classified into specific eukaryotic orthologous groups (KOG), 7259 genes were annotated with gene ontology (GO) terms, 9781 genes were mapped with the KEGG (Kyoto Encyclopedia of Genes and Genomes) ontology of 378 pathway maps. These annotation resources were used for the processing and quantification of RNA-seq.

**Table 2.** Statistics on gene model annotation of *M. hongkongensis*.

| Element | Value |
| --- | --- |
| Assembled genome size (bp) | 757,082,711 |
| Scaffold N50 (bp) | 72,332,161 |
| Number of Proteins | 40,086 |
| Number of longest Proteins only | 36,296 |
| Sum of Amino Acids (aa) | 15,406,827 |
| Mean of Proteins (aa) | 424 |
| Sum of Exons (bp) | 115,218,162 |
| Mean of Exons (bp) | 239 |
| Sum of Introns (bp) | 492,870,204 |
| Mean of Introns (bp) | 1201 |
| Numer of gene loci | 36,239 |
| Sum of gene region (bp) | 306,530,342 |
| % of gene loci in genome | 40.49% |
| Average gene region (bp) | 8459 |

The transcriptomes of ambient (25 °C) and heat shocked (32 °C) oysters were compared to understand how oysters responded to temperature changes. A total of 17,377 genes was found expressed in samples with the criteria of CPM over 1 in both replicates (Supplementary Table S13). Compared to the 25 °C group, in the heat-shocked group, 686 genes were up-regulated and 551 genes were down-regulated. Most differentially expressed genes got the $\log_2$(fold change) between $-5$ and 5 (Figure 3a).

The KOG enrich analysis showed the significant shift in gene usage in "signal transduction mechanisms" and "inorganic ion transport and metabolism" (Figure 3b). Top GO terms revealed that the functional gene group of metallopeptidases were up-regulated under heat shock. These genes are the protease enzymes that require metal for catalysis. Genes related to cell structure and cell junction formation were also up-regulated, while other genes on several metabolic processes, including lipid, fatty acid, inorganic acid, and carboxylic acid metabolism, were significantly down-regulated (Figure 3c). Further, 14 HSP genes were differentially expressed at 25 °C and 32 °C (Figure 3d).

KEGG annotation identified the significant up-regulation of genes in glutathione metabolism. Genes involved in the metabolic process from glutathione (GSH) to downstream metabolites were up-regulated, while the genes on the reverse process were down-regulated (Figure 3e). These results suggested that the glutathione content might drop, and other compounds would accumulate in the organism, including glutathione disulfide (GSSG), L-glutamate, glycine, and R-S-cysteine, which contribute to oxidative stress response [65,66].

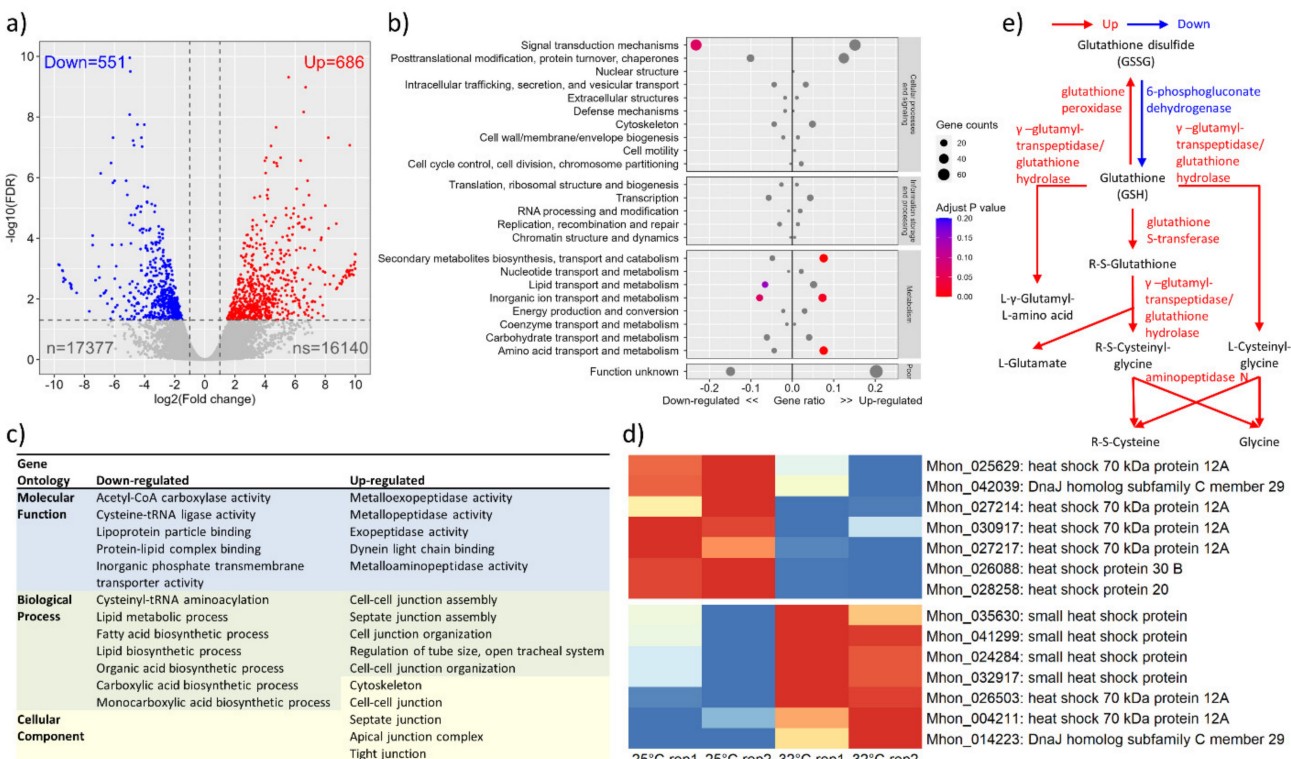

**Figure 3.** Transcriptome profiles of the oysters incubated under ambient (25 °C) and heat-shocked (32 °C) conditions. (**a**) Volcano plots showing distributions of DEGs; (**b**) KOG function enrich analysis; (**c**) top GO terms of DEG; (**d**) differentially expressed heat shock protein genes; (**e**) differential expression of genes involved in the glutathione metabolism under heat shock condition.

### 3.4. Alternative Splicing Events under Heat Shock Treatment

A total of 158 reliable alternative splicing events of 144 genes were identified in the transcriptomes with the FDR < 0.05 ( Supplementary Table S12). Percent spliced in (PSI) index was calculated for each event. At the level of |delta PSI| > 0.1, 58 AS and 92 AS showed preference on the inclusion of certain exon under heat shock and ordinary temperature, respectively (Figure 4a). Among all AS, 70 and 9 events were marked as cassette or multiple cassette exons, representing the exon skip events in the transcript, comprising 50% of all AS events (Figure 4b).

In 144 genes with AS, 109 genes had been annotated with KOG terms, representing three-fourths of the total number of genes (Figure 4c). The most abundant KOG term was "signal transduction mechanisms", and followed by "RNA processing and modification", with 24 and 18 genes assigned, respectively. Several protein kinases were recognised in the AS genes, including MAP kinase, cAMP-dependent protein kinase A, and casein kinase. Further, we uncovered that pre-mRNA processing factors and RNA splicing factors had also undergone AS and possessed isoforms, such as the LUC7-like protein, poly-U binding splicing factor and serine arginine-rich splicing factor. The KOG term of "posttranslational modification, protein turnover, chaperones" and "cytoskeleton" both had 13 genes annotated with specific function.

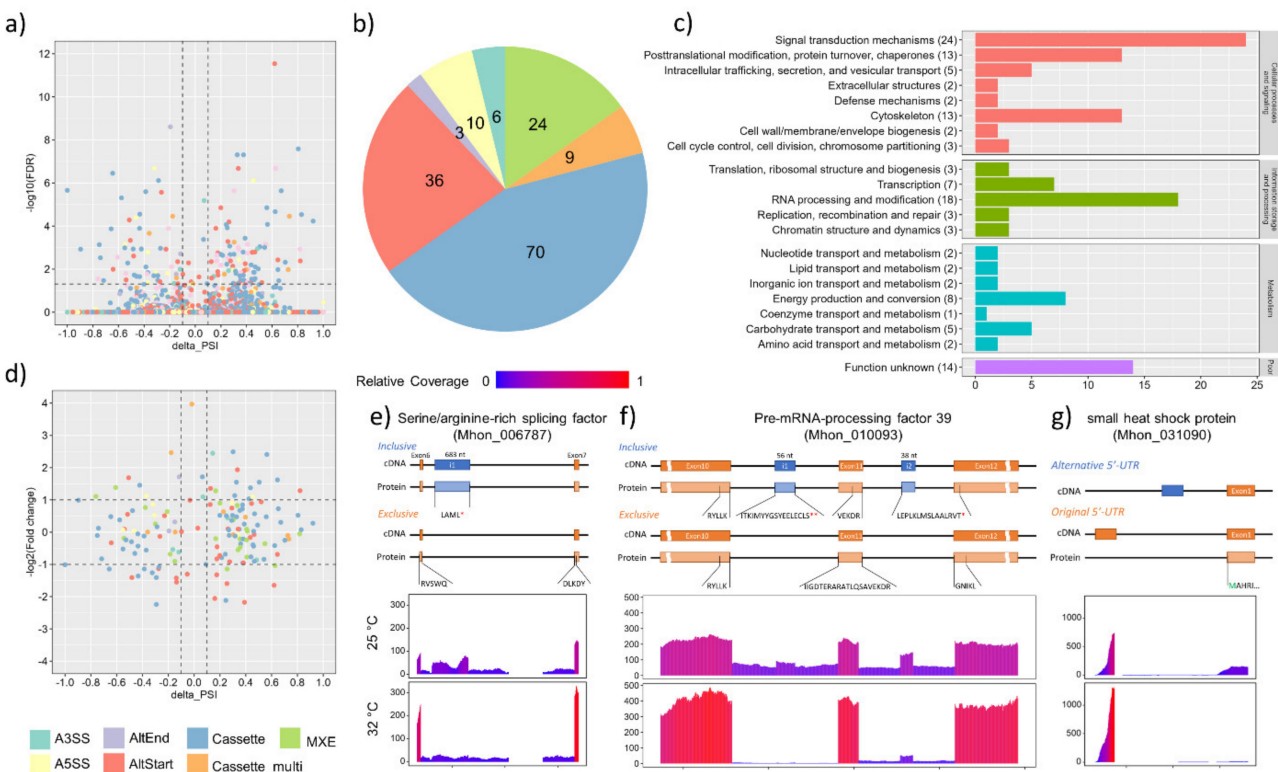

**Figure 4.** Alternative splicing (AS) events identified from the oysters incubated under ambient (25 °C) and heat-shocked (32 °C) conditions. (**a**) volcano plot showing the delta PSI value of AS candidates; (**b**) number and proportion of different AS types; (**c**) KOG annotation of genes with reliable AS prediction; (**d**) delta PSI value and gene expression fold change of AS genes under two incubation conditions; (**e**–**g**) alternative splicing patterns of: (**e**) serine/arginine-rich splicing factor Mhon_006787, (**f**) pre-mRNA-processing factor Mhon_010093, and (**g**) small heat shock protein Mhon_031090. Colour legend of AS types is the same for (**a**,**b**,**d**). Read count histogram of (**e**–**g**) is shaded by relative coverage from blue (low) to red (high).

Among those reliable AS identified in the transcriptome, eight and five of them were on genes up-regulated or down-regulated under heat shock, respectively (Figure 4d). This proportion showed no significant difference when compared to all expressed genes (Pearson's Chi-squared test = 0.14757, *p*-value = 0.7009). The AS on DEGs all had PSI change more than 0.1 between the ordinary and heat-shocked condition, except for a monocarboxylate transporter (Mhon_018589), where a multi-exon skip event occurred. Further, two up-regulated genes under heat shock with AS were annotated as serine/arginine-rich splicing factor (Mhon_006787) and pre-mRNA-processing factor 39 (Mhon_010093). Both of them possessed a higher level of exon-inclusive ratio at 25 °C. However, the inclusion of such exons resulted in the earlier termination of the protein sequence, resulting in misfunctioning (Figure 4e,f). In addition, a small heat shock protein (Mhon_031090) had an alternative start site at 25 °C, resulting in the variance of five prime untranslated regions (5'-UTR). Nevertheless, the mRNA amount of the alternative form was very little (Figure 4g).

### 3.5. Taxonomic Composition of Local Oyster Gut Microbiome

To clarify the taxonomic composition of gut microbiome in *M. hongkongensis* collected from Deep Bay, 16S rRNA gene amplicon of gut faecal samples from oysters were sequenced and clustered into OTUs ( Supplementary Table S13). A total of 322, 282, and 354 OTUs were identified with no less than 10 reads observed from three individuals, respectively (Figure 5a). Among them, 174 OTUs were shared among all three samples. At the phylum level, 11 phyla had an abundance of over 1% (Figure 5b). The top three taxonomies found

across the samples were Cyanobacteria, Proteobacteria and Spirochaetes, and they were all recognized as Gram-negative bacteria. Further, Brachyspiraceae (phylum Spirochaetes) and Synechococcaceae (phylum Cyanobacteria) were the two commonly identified families (Figure 5c). A total of six known bacterial genera with abundance over 1% were identified in the oyster gut samples (Figure 5d). Genus *Synechococcus* was found to be the most abundant richest one across all three individuals, with abundance varied from 2.75% to 8.88%. The genus *Mycoplasma*, a common symbiotic microbe, ranked at the second place with the abundance of 1.10–4.21%.

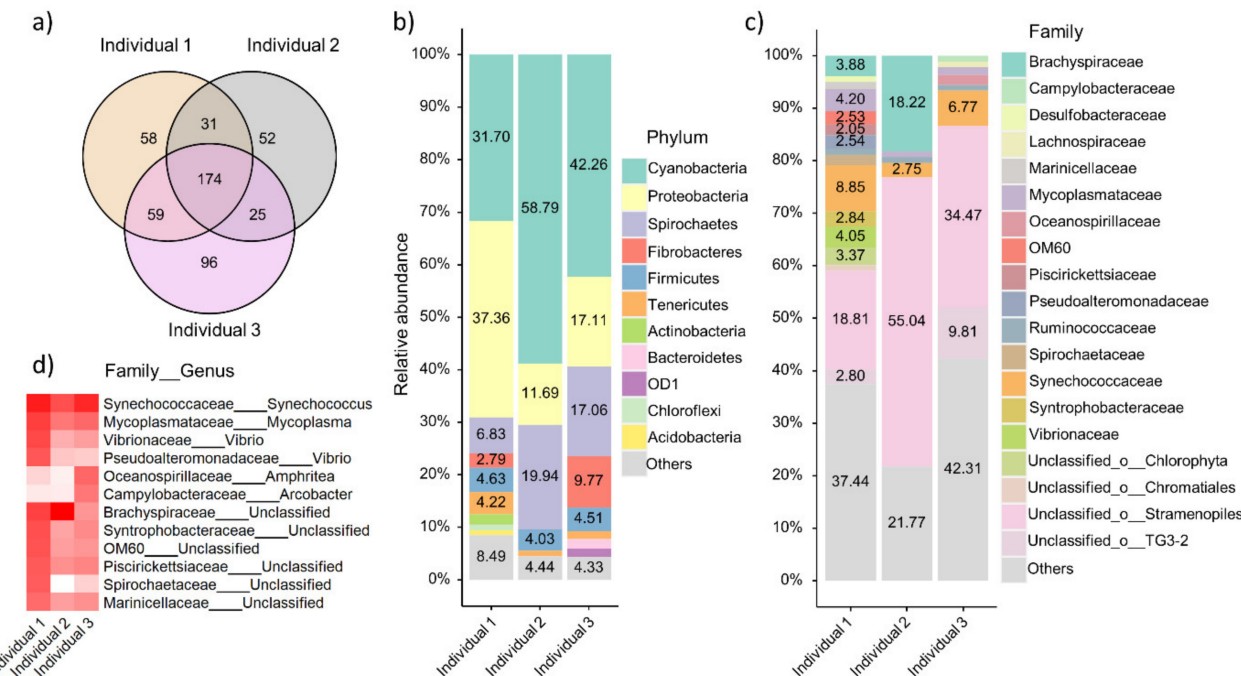

**Figure 5.** Gut microbiota diversity of three *M. hongkongensis* individuals. (**a**) Comparing OTUs identified in three oyster individuals; (**b**,**c**) phylum and family abundance of the common OTUs—only OTUs with an abundance over 1% are listed and abundance over 2% are labelled on chart; and (**d**) comparison on the abundance of common genera, white to red (low to high)—OTUs with an abundance over 1% in at least one individual.

## 4. Discussion

Deep Bay and Lantau Island share similar environmental conditions, and are separated from each other by approximately 25 km. Our whole genome sequencing results indicate the close relationship across the local *M. hongkongensis* oysters. At a broader geographical scale, *M. hongkongensis* sampled along about 1000 km of the coast of southern China (Fujian, Guangdong and Guangxi provinces), showed a significant population structure geographically, as indicated by the microsatellite loci and COI gene [21,67]. In the estuarine oyster *Magallana ariakensis*, Pacific oyster *M. gigas*, and Olympia oyster *Ostrea lurida*, populations and gene flow barriers could also be identified [20,29,68]. Future analyses on the population structure of *M. hongkongensis* across a wider geographical scale are needed, as hatchery and farming activities have been identified as influential in determining oyster population structure [69,70], and useful for the identification of novel targets for molecular breeding [71,72].

Oysters inhabit the highly changeable river estuarine-coastal environments, which are charisterised by dynamic temperature, salinity, pH, and nutrient conditions [73–75]. Previous studies comparing the genomes of oysters inhabiting warm and low-salinity (e.g., *M. ariakensis*, *C. virginica* and *M. hongkongensis*) and cool and high-salinity (e.g., *M. gigas*) estuarine environments, reveals significant expansion of specific gene families. These gene families include heat shock proteins and solute carrier proteins which have been suggested

to be the genetic bases of enhanced stress adaptation plasticity for oysters that live in warm and low-salinity environments [29]. Here, we observed that site-specific genetic variants in the homeobox and HSP genes are essential regulators of body development and stress responses. As genetic divergence and its associated functional changes could strongly diversify the adaptive regulations and responses of oysters to environmental factors [28,76–78], it is possible that the variance of environmental condition or source of oyster recruits between Shenzhen Bay and Lantau Island might lead to the possession of these genetic variants which warrants further investigation.

The transcriptomic results reported here reveal protein binding genes and signal transduction genes as the major groups of genes that respond to temperature changes. These functional groups underwent expansion during the evolution of oysters, and they possess sophisticated stress adaptation mechanisms at both the genomic and transcriptomic level [28,29,79]. Previous studies suggest that HSP might be one of the central defense mechanisms for several stressors in oysters, including salinity, heavy metals, and air exposure [28]. Genetic manipulations on these signaling pathway genes could potentially provide stress-tolerant oysters [80,81].

Alternative splicing (AS) on HSP and alternative oxidase transcripts play roles in stress responses in the oysters *M. gigas* and *C. virginica* [82,83]. In the Pacific oyster, *M. gigas*, AS on over 2000 genes were shared by different stressors, with functional enrichment on signal transduction regulation and post-translational protein modification [82]. In our case, we only identified 144 AS in *M. hongkongensis* transcriptome, however, we detected heat-stress induced expression accumulation and AS on two splicing factors in *M. hongkongensis*. These AS results suggest that mRNA with an early stop codon may subsequently lead to nonsense mRNA decay rather than protein production [84]. Such AS splicing and disruption of protein production has been found in other organisms in response to various environmental stressors [85–87]. AS splicing may, therefore, represent a widely-used response to environmental stress.

Gut microbiota is an important modulator for the development and stress responses of the hosts [88,89]. Our analyses profiled the microbiota in the guts of *M. hongkongensis* from Hong Kong, which showed similar profiling to that of pearl oysters (*P. f. martensii*) and Pacific oysters (*M. gigas*) collected from Guangdong Province, where the phyla Cyanobacteria, Proteobacteria, and Tenericutes were found to be dominant in the microbial communities [38,90]. In response to environmental changes, the microbe community in the gut can transform and play a part in host immunometabolism [91–97]. For instance, gut microbiota was found to regulate the glutathione metabolism in mice and changed the oxidative stress response [98,99]. In our transcriptome profiles, differential regulation of glutathione metabolism during heat shock was also observed. Given heat stress might cause change to the redox equilibrium [100,101], depletion of glutathione can further activate the transcription of heat shock genes, regulating the global stress responses [102,103]. Further analyses on the differences of the microbiome in oysters under heat shock are needed to validate the functional roles of microbiota on stress response via regulation of the host glutathione metabolism.

## 5. Conclusions

Adaptation to a changeable environment is important for the survival and development of oysters. Here, we analysed the population structure of the oyster *M. hongkongensis* by whole genome re-sequencing and identified genetic variants in homeobox and heat shock protein genes. We revealed the transcriptional regulation and response to heat shock. The gut microbiota composition of oysters in the area was also characterised. Together, this study provides a comprehensive understanding of the heat stress response and regulatory mechanisms of *M. hongkongensis* and sheds light on the local adaptation process in changeable estuarine–coastal environments.

**Supplementary Materials:** The following supporting information can be downloaded at: https://www.mdpi.com/article/10.3390/jmse10020237/s1: data spreadsheets of Supplementary Tables S1–S13. Table S1: GPS coordinates of oyster samples; Table S2: Heat shock proteins annotated in *M. hongkongensis*; Table S3: Primer sequences for SNP validation of ScQ1DuM_12504:36310998 in 3 individuals collected at B7; Table S4: Genome sequencing data; Table S5: Transcriptome sequencing data; Table S6: SNPs and INDELs of Homeobox genes among 49 oyster samples; Table S7: SNPs and INDELs of Hsp70 genes among 49 oyster samples; Table S8: Unique SNPs and INDELs of Hsp70 genes among collection batches; Table S9: Unique SNPs and INDELs of Homeobox genes between Lantau (HKU) and Shenzhen Bay (SZB); Table S10: Unique SNPs and INDELs of Hsp70 genes between Lantau (HKU) and Shenzhen Bay (SZB); Table S11: Gene expression level under normal (25 °C) and heat shock (32 °C) conditions; Table S12: Alternative splicing prediction under normal (25 °C) and heat shock (32 °C); Table S13: OTU classification of 16S amplicons of oyster gut microbiome.

**Author Contributions:** Conceptualization, J.H.L.H.; data curation, Y.X., W.N.; formal analysis, Y.X., W.N., S.T.S.L., Y.Y., and Y.L.; funding acquisition, J.H.L.H.; B.D.R.; investigation, E.Y.Y.H., S.T.S.L., K.C., Y.Y., Y.L., C.F.W., H.Y.Y., and P.W.S.J.; field collection, J.H.L.H., W.N., H.Y.Y., K.C., and K.M.C.; writing, Y.X., E.Y.Y.H., K.C., K.M.C., K.H.C., B.D.R., L.J.F., and J.H.L.H. All authors have read and agreed to the published version of the manuscript.

**Funding:** This research was funded by Hong Kong Research Grants Council Collaborative Research Fund (C4015-20EF), the Open Collaborative Research Fund and Operation Fund from the Southern Marine Science and Engineering Guangdong Laboratory (Guangzhou) (HKB L20200008), The Chinese University of Hong Kong Direct Grant (4053433, 4053489), and the Marine Ecology Enhancement Fund from Airport Authority Hong Kong (MEEF2019008).

**Acknowledgments:** The authors would like to thank Satya Narayana for discussion.

**Conflicts of Interest:** The authors declare no conflict of interest.

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
