# Peer review of "Population Genomics, Transcriptional Response to Heat Shock, and Gut Microbiota of the Hong Kong Oyster Magallana hongkongensis"

_jmse, doi:10.3390/jmse10020237_

Round 1
Reviewer 1 Report
Yichun Xie et al. performed a population structure analysis of M. hongkongensis oysters from Deep Bay and Lantau Island in Hong Kong. The author used genomic approaches and strong statistical analysis skills. This work mainly reveals the molecular basis for the adaptation of the oyster M. hongkongensis to environmental conditions. This work represents an important resource for Hong Kong seafood production. Indeed, Hong Kong is a big consummer of seafood.
However, I have some minors concerns I would like to see addressed befor publishing this paper.
First, I wan to suggest authors, to highlight in their introduction, the importance of seafood in Hong Kong and in particular that of the oyster. For example, Hong Kong imported in 2018, about 124,000 tons of seafood, including an important part of oyster. This makes it possible to consolidate the idea of ​​studying such a species.
Paragraph 2.2 : This paragraph must be splited in two short paragraphs. The first must includde from line 126 to line 138 and can be titled « Bioinformatic analisis ». The second one must include line 139 to line 143 and can the second paragraph can keep the initial name of 2.2.
Paragraph 2.2 – Line 130 : The BWA algorithm is need
Paragraph 2.2 – Line 133 : The GATK algorithm is need
Author Response
Reviewer 1 comments 1) "First, I wan to suggest authors, to highlight in their introduction, the importance of seafood in Hong Kong and in particular that of the oyster. For example, Hong Kong imported in 2018, about 124,000 tons of seafood, including an important part of oyster. This makes it possible to consolidate the idea of studying such a species." Response 1: We have now added the following sentences in the introduction: "Hong Kong has one of the top seafood consumers per capita in the world" 2) "Paragraph 2.2 : This paragraph must be splited in two short paragraphs. The first must includde from line 126 to line 138 and can be titled « Bioinformatic analisis ». The second one must include line 139 to line 143 and can the second paragraph can keep the initial name of 2.2." Response 2: The paragraphs are separated as suggested. 3) "Paragraph 2.2 – Line 130 : The BWA algorithm is need" Response 3: The following information is now included: "The processed reads were then mapped to the published Hong Kong oyster M. hongkongensis genome [12] using Burrows-Wheeler Aligner (BWA) (version 0.7.12) with “-M” activated (mark shorter split hits as secondary) and other parameters on default values, for further SNPs analysis of single nucleotide polymorphisms (SNPs). PCR duplicates were removed with Picard ‘MarkDuplicates’." 4) "Paragraph 2.2 – Line 133 : The GATK algorithm is need" Response 4: The following information is now added: "SNPs were called with the ‘VariantFiltration’ function in Genome Analysis Toolkit (GATK) with hard filtering criteria (--filter-expression “FS > 60.0 || MQ < 40.0 || QD < 2.0 || SOR > 3.0”) [44]."Reviewer 2 Report
This manuscript describes the genomic, transcriptomic, and microbiome diversity of the Hong Kong oyster. The experiments were carefully done ensuring that the samples analyzed were indeed M. hongkongensis. I believe the analysis presented would be of interest to a wide variety of shellfish researchers.
Comments:
Were the oyster collected in the intertidal? In reefs? Attached to something? In sediment?
Indicate the coverage for the WGS samples. From the supplementary file, it looks like the coverage was 3-6X. Isn't this coverage problematic in identifying heterozygous loci?
Why is Table 2 in the manuscript? Hasn't the genome you are using been already annotated [ref 12]? What was problematic with the annotation in ref 12?
Author Response
Reviewer 2 comments 5) "Were the oyster collected in the intertidal? In reefs? Attached to something? In sediment?" Response 5: The following information is now added: "A total of 44 M. hongkongensis samples were collected from oyster rafts at different localities of Deep Bay (Shenzhen Bay) in May and June 2019 (named as B1-B10 and KM), and five additional samples were collected from intertidal mudflat at Lantau Island in September 2019 (named as HKU) (Fig. 1 and Supplementary information 1, Table 1)." 6) "Indicate the coverage for the WGS samples. From the supplementary file, it looks like the coverage was 3-6X. Isn't this coverage problematic in identifying heterozygous loci?" Response 6: We agree with the reviewer. While this range of coverage could underestimate the heterozygosity of loci, the variant discovery is recovered by collecting 2-5 biological replicates from each locality or batch of samples.7) "Why is Table 2 in the manuscript? Hasn't the genome you are using been already annotated [ref 12]? What was problematic with the annotation in ref 12?"
Response 7:
There is nothing wrong or problematic with the annotation in reference 12. We showed the information for clarity from the readers point of view. If the reviewer would prefer, we can also take out the Table 2 away from the manuscript.